# Repurposing Disulfiram as an Antifungal Agent: Development of a New Disulfiram Vaginal Mucoadhesive Gel

**DOI:** 10.3390/pharmaceutics15051436

**Published:** 2023-05-08

**Authors:** Maria Lajarin-Reinares, Iria Naveira-Souto, Mireia Mallandrich, Joaquim Suñer-Carbó, Montserrat Llagostera Casas, Maria Angels Calvo, Francisco Fernandez-Campos

**Affiliations:** 1Department of Genetics and Microbiology, Campus Microbiology Unit, Autonomous University of Barcelona, 08193 Bellaterra, Spain; montserrat.llagostera@uab.cat; 2R & D Development, Reig Jofre Laboratories, 08970 Sant Joan Despí, Spain; inaveira@reigjofre.com; 3Department of Pharmacy and Pharmaceutical Technology and Physical Chemistry, School of Pharmacy and Food Sciences, University of Barcelona, 08028 Barcelona, Spain; mireia.mallandrich@ub.edu (M.M.); jsuner@ub.edu (J.S.-C.); 4Department of Animal Health and Anatomy, Faculty of Veterinary, Autonomous University of Barcelona, 08193 Bellaterra, Spain; mariangels.calvo@uab.cat; 5R & D Department, Labiana Pharmaceuticals, 08757 Corbera Llobregat, Spain; francisco.fernandez@labiana.com

**Keywords:** disulfiram, *Candida* spp., mucoadhesion, ATP-binding cassette, resistant, vaginitis

## Abstract

Alternative formulations need to be developed to improve the efficacy of treatments administered via the vaginal route. Mucoadhesive gels with disulfiram, a molecule that was originally approved as an antialcoholism drug, offer an attractive alternative to treat vaginal candidiasis. The aim of the current study was to develop and optimize a mucoadhesive drug delivery system for the local administration of disulfiram. Such formulations were composed of polyethylene glycol and carrageenan to improve the mucoadhesive and mechanical properties and to prolong the residence time in the vaginal cavity. Microdilution susceptibility testing showed that these gels had antifungal activity against *Candida albicans*, *Candida parapsilosis*, and *Nakaseomyces glabratus*. The physicochemical properties of the gels were characterized, and the in vitro release and permeation profiles were investigated with vertical diffusion Franz cells. After quantification, it was determined that the amount of the drug retained in the pig vaginal epithelium was sufficient to treat candidiasis infection. Together, our findings suggest that mucoadhesive disulfiram gels have the potential to be an effective alternative treatment for vaginal candidiasis.

## 1. Introduction

Vaginal candidiasis (VC), often referred to as vulvovaginal candidiasis, is mainly caused by the *Candida* species and is the second most prevalent infection of the vaginal mucosa after bacterial vaginosis [1]. VC affects approximately 75% of women of reproductive age at least once during their lives, and it is estimated that 40–50% will experience recurrence and 8% will develop chronic VC [2,3,4].

*Candida* spp. are commensal fungi in the human microflora that are found in the lower genital tract in 10–20% of healthy women. Although part of the normal microflora, *Candida* can become opportunistic agents with a tendency to overgrow [2]. The most common *Candida* species involved in VC is *C. albicans* (85–90%), followed by *Nakaseomyces glabratus* (*C. glabrata*), *C. tropicalis*, *Pichia kudriavzevii* (*C. krusei*), and *C. parapsilosis* [5]. Approximately 25 to 40% of women who are culture-positive for *Candida* are asymptomatic. The natural evolution of asymptomatic colonization to symptomatic infection is not clear. There are multiple mechanisms that contribute to the colonization of *Candida* and its symptomatic infection, such as genetic host factors (polymorphisms in the blood group), hormones, the use of antibacterial agents, age, sexual activity, certain pathologies (diabetes mellitus and immunodepression), idiopathic causes, pregnancy, and an imbalance in the vaginal microbiota [4,6,7]. The most common clinical manifestations of VC are pruritus, hyperemia, vaginal discomfort, leucorrhea, burning, soreness, dyspareunia, and vaginal erythema [7]. Furthermore, colonization of the vaginal epithelium contributes to the deterioration of mucosal surfaces, facilitates the formation of yeast reservoirs, or becomes a source for future infections, leading to chronic infections and serious consequences (i.e., infertility and sterility) [8].

The treatment of noncomplicated VC involves the use of short-term local therapy (up to 3–7 days) or single-dose oral antimycotics, mainly those in the azole family, which is effective in 90% of cases. The most common locally applied azoles are clotrimazole, butoconazole, and miconazole. Fluconazole is usually given orally between 1 and 3 days [9]. Topical azoles may provide more immediate relief; however, some patients experience hypersensitivity, local itching, or burning [10]. For vaginal drug delivery, several pharmaceutical formulations are available for consumers, including creams, ovules, tablets, and gels [8,11,12]. These treatments exhibit some drawbacks, such as low permeation in the underlying epithelium (to remove reservoirs) and low residence time owing to the self-cleaning action of the vaginal tract. (It is recommended to be administered at night before lying in bed.) These factors could lead to low drug exposure and the development of resistance. Many studies have documented the ability of *Candida* to develop high-level resistance to antifungal therapy [13,14,15,16]. Resistance mechanisms include overexpression of or mutations to the target enzyme of azoles, lanosterol 14a-demethylase, and overexpression of ATP-binding cassette (ABC) efflux proteins [17]. Among ABC transporters, Cdr1p plays a key role in azole resistance in *C. albicans* [18].

Disulfiram (D) is a dithiocarbamate used to treat alcoholism that was approved by the US Food and Drug Administration (FDA) in 1951. D produces discomfort in patients who drink alcohol as a strategy for their dishabituation, inhibiting aldehyde dehydrogenase (ALDH), which results in the specific accumulation of acetaldehyde [19]. In addition, D is characterized by a low molecular weight (296 Da), logP value of 3.88, melting point under 200 °C (71.5 °C), and fewer than five hydrogen bond donors and acceptors with a strong ability to chelate metals [20]. In accordance with its physicochemical properties, a good permeation profile is expected. Different in vitro and in vivo studies have shown that D could be an active substance against scabies, lice, and bacteria [21,22]. Furthermore, the effectiveness of D against fungi and cancer was reported [23,24]. Different mechanisms of action have been proposed to explain the antimicrobial activity of D: metal ion chelation, which affects cell homeostasis, and the presence of reactive thiol groups, which can impact the functions of proteins by forming incorrect disulfide bonds [22]. D was found to be potentially useful by inhibiting the function of glycoprotein P (P-gp) and has also been shown to act as a modulator of other multidrug transporters, such as MRP1 and MRP4 [25]. Because of the functional similarities between Cdr1p and human P-gp, D is a potent modulator of Cdr1p [17]. Therefore, D is an attractive agent to develop for candidiasis treatment.

The aim of this work is to develop and optimize a mucoadhesive drug delivery system for the local administration of D. The in vitro antifungal efficacy of D was studied with different *Candida* species, and the mucoadhesion of the formulation was enhanced using polymers with different adhesive mechanisms. The proposed formulation represents an alternative to the commercially available treatments for VC due to the selected novel drug and mucoadhesive polymers that would increase the residence time of the treatment and yeast drug exposure.

## 2. Materials and Methods

### 2.1. Materials

Sodium chloride (0.35% *w*/*v*) (Quimivita, Barcelona, Spain), potassium hydroxide (0.14% *w*/*v*), calcium hydroxide (0.02% *w*/*v*), acetic acid (0.1% *w*/*v*) (Scharlab, S.L., Sentmenat, Spain), bovine serum albumin 0.002% (*w*/*v*) (Merck Life Science, S.L.U., Madrid, Spain), lactic acid (0.2% *w*/*v*) (Escuder, S.L., Rubi, Spain), glycerol 0.02% (*w*/*v*) (Caila & Pares, S.A., Barcelona, Spain), urea (0.04% *w*/*v*) (Quality Chemicals, S.L., Esparraguera, Spain) and glucose (0.5% *w*/*v*) (Quimidroga, S.A., Barcelona, Spain) were used to prepare simulated vaginal fluid (SVF) [26].

Mucoadhesive gels were produced with polyethylene glycol (PEG-90M) (Colorcon Limited, Kent, UK) and carrageenan (DuPont, Wilmington, DE, USA); benzylic alcohol (Scharlab, S.L., Sentmenat, Spain) was chosen as a preservative; and hydroxypropyl-beta-cyclodextrin (HPCD) (Pracofar, Martorelles, Spain) was used as an active solubilizing agent. D (Farchemia S.r.l., Treviglio, Italy) was the active pharmaceutical ingredient, and the selected solvent was water purified in-house. Sodium fluorescein (Scharlau, Barcelona, Spain) was used in the mucoadhesive study as a marker.

### 2.2. Antifungal Susceptibility Testing

The fungal strains used in this study were purchased from the Spanish Collection of Type Cultures (CECT). *C. albicans* CECT 1002 (which corresponds to ATCC 18804), *C. parapsilosis* CECT 10434 (MCYC 2031), and *C. glabrata* CECT 1448 (ATCC 2001) were used for the antifungal susceptibility test.

The minimum inhibitory concentration (MIC) of D was determined by the broth microdilution method in 96-well microplates according to guidelines M27-A3 of the Clinical and Laboratory Standards Institute (CLSI) standard [27]. A 24 h culture of *Candida* adjusted to a 0.5 McFarland was used to prepare the initial inoculum. A working suspension was generated by making a 1:100 dilution followed by a 1:20 dilution of the stock suspension with RPMI 1640 broth medium buffered with MOPS (3-(N morpholino)propanesulfonic acid) (Thermo Fisher Scientific, Waltham, MA, USA) at pH 7.0. Stock solutions (10 mg/mL) of D were prepared in ethanol, and D was tested from 64 (the highest soluble concentration) to 0.125 µg/mL. The final ethanol concentration was below 0.5% to not alter yeast growth. The negative control was the respective medium, and a control with the same ethanol concentration as that in the samples with D was evaluated. Furthermore, each fungal strain in medium was used as a positive control, and amphotericin B was used as a positive control of inhibition. The plates were incubated at 35 °C for 48 h, and then, the MIC values were determined. The M27 methodology gave reproducible results when testing *Candida*, and the establishment of interpretive breakpoints has begun. However, it is becoming clear that the M27 methodology may be inadequate for certain organisms and antifungal agents. For example, determining the susceptibility of *Candida* isolates to amphotericin B may be a concern [28]. D is not described in these guidelines, and the suitability of the M27 method is not known. To resolve the inherent possible variability of MIC determination, MICs were also determined an alternative medium (Sabouraud dextrose broth).

### 2.3. Preparation and Optimization of the Mucoadhesive Gel

For the mucoadhesive placebo gel preparation, purified water (q.s., 100% *w*/*w*) was heated to 50 °C with HPCD (20% *w*/*w*). PEG-90M was added, it was stirred at 750 rpm with a mixing paddle, and then stirring was reduced to 150 rpm until complete dissolution. Finally, carrageenan was added with stirring at 75 rpm until complete homogenization. Disulfiram gels were produced in the same way by adding D (0.5% or 1% *w*/*w*) to purified water with HPCD. Fluorescein (0.01% *w*/*w*) mucoadhesive gels were produced to study the in vitro mucoadhesive properties.

The properties of formulations were influenced by different experimental variables, such as the amounts of the polymers. To obtain the most suitable mucoadhesive gel for vaginal application, the influence of the concentrations of PEG-90M and carrageenan on the dependent variables (mucoadhesion, fluorescein adhesion, and viscosity) was studied (Table 1) with placebo formulations. Central composite design was employed to analyze the main and combined effects of these variables on the responses. For this, 5 central points were selected, and a default α value of 1.41 was used to determine the axial points. A significance level (α) of 0.05 was established. The choice of the model was made based on the values of R^2^, adjusted R^2^, and AIC (Akaike information criterion). In addition, the production order of the formulations was randomized to reduce the chances of external design factors affecting our results.

Statistical analysis of the variables studied was carried out using Minitab 17 statistical software (Minitab, Inc., 2010, State College, PA, USA) to obtain mathematical equations for each model. Then, optimization of each model was performed (Section 2.7).

### 2.4. Adhesive Strength of the Gels

The adhesive strengths of the placebo gels were evaluated by modifying the devices from Mei et al. [11] (Figure 1). To simulate the vaginal epithelium, agar hydrogels (4% *w*/*w*) (Scharlab, Barcelona, Spain) were prepared with type II pig mucin (0.34% *w*/*w*) (Sigma–Aldrich, St. Louis, MI, USA) [29]. This gel was attached to two planks, as shown in Figure 1. The formulation (0.5 g) was placed on the bottom plank, and to evaluate the effect of vaginal fluid on the formulations, SVF (0.25 mL) was applied to the formulation before the upper plank was pressed onto the bottom plank and the formulation.

One milliliter of water was added every 15 s to a tared beaker fixed on the opposite side of the balance. Once the two planks were separated, the weight of the water (W_1_) was used to calculate the tensile stress (σ; mN/cm^2^) according to Equation (1):σ = (W_1_ × *g*)/*s*(1)
where *g* is gravity (m/s^2^), and *s* is the agar surface (cm^2^).

### 2.5. Viscosity of the Mucoadhesive Gels

As a screening step in the selection of the final formulation composition, the viscosities of the mucoadhesive gels were evaluated using a rotatory viscosimeter (Brookfield, Manassas, VA, USA). The viscosities of the samples were measured 24 h after production of each formulation. Moreover, viscosities of the same samples with SVF in a ratio of 1:6.67 (SVF:gel) were measured after mixing and resting for 48 h. Ten milliliters of each sample was applied to the sample container at 25.0 ± 0.5 °C. Viscosity measurements were conducted for 30 min at 10 rpm (spindle SC4-29). The ratio of SVF to gel used was considered to allow approximately 0.5–0.75 mL of SVF to be present at any one time [30].

### 2.6. Mucoadhesive Properties

For the mucoadhesive studies, a vagina and its physiological conditions were simulated using a tube with a portion of agar (8 × 2 cm) with 0.34% *w*/*w* mucin, on which 1 g of mucoadhesive gel with fluorescein (0.01% *w*/*w*) was placed on top of the tube. The tubes were mounted at an approximate angle of 70° under physiological conditions. SVF at 1 rpm was passed through the tube using a peristaltic pump (Shenchen Pump YZ1515x, Hebei, China) at 0.0133 mL/s, the lowest flow of the peristaltic pump. Friedl et al. used a total of 6 mL of SVF over 24 h (*v* = 0.00006944 mL/s), taking samples at 4 h, 8 h, 12 h, and 24 h [31]. Extrapolating these sampling times to the flow of our system, the determined sampling times were as follows: 1.15 min, 2.30 min, 3.45 min, and 7.30 min. The whole experimental setup is illustrated in Figure 2.

At the end of the study, the concentration of fluorescein released (F_2_) at each time point was determined by measuring the fluorescence at a wavelength of 485 nm using a Victor Nivo multimode plate reader (Perkin Elmer, Waltham, MA, USA). Before sample measurement, a fluorescein calibration curve was prepared by measuring the fluorescence of fluorescein solutions in water from 7.6 × 10^−4^ μg/mL to 2 × 10^−5^ μg/mL (R^2^ = 0.999928).

Once the samples were quantified, the fluorescent dose remaining in the agar (F_3_) was determined by calculating the difference from the fluorescein dose initially seeded (F_1_) following the equation
F_3_ = F_2_ − F_1_

The F_3_ value was used as an indirect parameter of the mucoadhesion of the formulation.

### 2.7. Optimization of the Mucoadhesive Gels

Once the mathematical models for each variable were obtained, the two factors (PEG-90M and carrageenan concentration) were examined to optimize the mucoadhesive properties and viscosity of the formulation. The concentration of polymers giving the formulation the highest mucoadhesion and lowest viscosity was determined.

The mucoadhesive properties and viscosity of the optimized final formulation placebo and the formulations with 0.5 and 1% D (*w*/*w*) were studied. Furthermore, from the mathematical model obtained during the statistical study, the bias was calculated (Equation (2)) to determine the predictive capacity of the model.
(2)$$\%\mathrm{Bias}=\frac{\left|Ref-Z_{1}\right|}{Ref}\times 100$$
where *Ref* is the theoretical reference value of the model, and Z_1_ is the mean of the experimental value obtained.

Additionally, the rheological properties (Section 2.8) and syringeability (Section 2.9) of the placebo-optimized formulation and the release (Section 2.10) and permeation (Section 2.11) profile of the final D formulation were characterized.

### 2.8. Final Rheological Properties

Placebo-optimized formulation rheology, thixotropy, and viscoelasticity measurements were made by performing rotational and oscillatory tests using a Haake Rheostress^®^ 1 rheometer (Thermo Fisher, Karlsruhe, Germany).

Rotational measurements. Steady-state measurements were made with a parallel pate and plate geometry (P35Ti L: 60 mm diameter, 2° angle). The shear stress (τ) was measured as a function of the shear rate ($\dot{\gamma }$). Viscosity curves (η = f(γ)) and flow curves (τ = f($\dot{\gamma }$)) were recorded at 37 ± 0.1 °C. The shear rate ramp program included a 3 min ramp-up period from 0 to 100 s^−1^, a 1 min constant shear rate period at 100 s^−1^, and a 3 min ramp-down period from 100 to 0 s^−1^. Representative mathematical models were fit to the flow curves to search for the best descriptive model (Table 2) [32]. Selection of the best-fitting model was based on the correlation coefficient (observed vs. predicted) and Chi-square value. The apparent thixotropy (Pa/s) was estimated as the area of the hysteresis loop. Steady-state viscosity (η, Pa s) was determined from the constant shear section at 100 s^−1^.

Oscillatory test. The oscillatory test was performed with parallel plate–plate geometry (P35Ti L: 60 mm diameter, 2° angle). A strain sweep test was performed with a strain range of 0.01 to 500 Pa and an oscillation rate of 1 s^−1^ to determine the linear viscoelastic region (LVR) of the samples. During each sweep stress test, the changes in the storage and loss moduli and the phase angle (G′, G″, and δ, respectively) were plotted as a function of shear stress. Afterward, a frequency sweep test was carried out between 0.1 and 10 s^−1^ at a constant shear rate within the LVR to determine the related variations in the G′, G″, and complex viscosity (η*), which were used for sample characterization.

All obtained data were analyzed with HAAKE RheoWin^®^ Data Manager v. 4.88.

### 2.9. Syringeability of the Formulations

Syringeability was determined by calculating the time needed for 5 g of the product to fall from a cannula with the application of a constant force of 9.688 N (applied weight of 987.6 g) [33].

### 2.10. In Vitro Release Tests of the Disulfiram Gel

The in vitro release of D from the mucoadhesive gels was studied using vertical Franz cells (Vidrafoc, Barcelona, Spain) with a 12 mL receptor compartment and an effective diffusion area of 1.54 cm^2^. A 15% *w*/*w* HPCD solution in PBS (pH 5.5) was used as the receptor medium (RM) at 37 °C with stirring at 500 rpm to maintain sink conditions throughout the experiment due to the poor water solubility of D (4.09 mg/L) [20] and according to previous studies reported [21,22]. A total of 0.0545 g/cm^2^ gel was applied in the donor compartment, corresponding to 420 μg of D for the 0.5% formulation (*n* = 5) and 840 μg of D for the 1% formulation (*n* = 5). A membrane with a 0.45 μm pore diameter was used (Supor PES membrane, Thermo Fisher, Waltham, MA, USA). Aliquots of 300 µL were taken at certain times (0, 0.5, 1, 1.5, 3, 4, 5, and 6 h). The samples were analyzed using High-Performance Liquid Chromatography (HPLC) with a method previously reported for similar studies [21] to quantify the amount of D that had diffused through the membrane.

Kinetic modeling of the release data was carried out with the DD-solver Excel Add-on [34]. The mean values of the release curve were adjusted to nonlinear models, i.e., first-order, Higuchi, Korsmeyer–Peppas, and Weibull (Table 3), to select the population behavior. Then, individual data were adjusted according to the selected population model. Model selection was based on the AIC, which reflected the lowest deviation of the model with respect to the empirical data [35]. The mean and standard deviation of the parameters were reported.

In Table 3, if *n* is between 0.43 and 0.85, then the release mechanism follows an anomalous transport mechanism. In the case of *β*, for values lower than 0.75, the release follows Fickian diffusion, either in Euclidian (0.69 < *β* < 0.75) or fractal (*β* < 0.69) spaces. Values (*β*) in the range of 0.75–1.0 indicate a combined mechanism, which is frequently encountered in release studies [36].

### 2.11. Pig Vagina Permeation

Pig vaginas were obtained at the time of sacrifice from a local abattoir (Barcelona, Spain). The full thickness vaginal mucosa (approximately 6.2 mm) was carefully debrided, cleaned with sterile saline solution, and transported to the laboratory at 4 °C in saline solution. For this study, tissues were frozen by placing them in containers with a PBS mixture containing 4% albumin and 10% DMSO (as cryoprotective agents) and stored (for a maximum of 1 month) at −80 °C in a mechanical freezer. DMSO produces adverse effects at room temperature; therefore, the addition of DMSO prior to freezing was performed at 4 °C, whilst thawing involved immersion in a water bath filled with PBS at 37 ± 1 °C and gentle shaking for 30 min, until total elimination of DMSO was achieved [37]. On the day of the experiment, the vaginal pieces were thawed and mounted on Franz cells (*n* = 4 for each dose) with an effective diffusion area of 0.64 cm^2^ and approximately 5 mL of receptor volume capacity (HPCD 15% in PBS, pH = 5.5 with stirring at 600 rpm). A total of 0.0545 g/cm^2^ gel (0.5% or 1%) was administered in infinite doses under nonoccluded conditions. Samples from the receptor compartment (300 µL) were taken at regular time intervals of 1.5, 2.5, 3.5, 4.5, and 6 h. Samples were analyzed and quantified using the HPLC method described in the previous section.

After drug quantification, the following permeation parameters were calculated: transmucosal flux (*J*, μg/cm^2^ h) (Equation (3)), permeability coefficient (*Kp*, cm/h) (Equation (4)), lag time (*T_lag_*, h) (obtained by linear extrapolation of the *x*-axis of the points at steady state), diffusion parameter (*Dif*, 1/h) (Equation (5)), and partition parameter (*P*, cm/h^2^) (Equation (6)).
(3)$$J=\frac{dQ}{dT\times S},$$
where *J* is the transdermal flux, *dQ* is the difference in the amount permeated, *dT* is the time differential, and *S* is the membrane diffusion surface.
(4)$$Kp=\frac{J}{C_{d}},$$
where *C_d_* (μg/mL) is the concentration of the drug in the donor compartment.
(5)$$Dif=\frac{1}{{6T}_{lag}}.$$
(6)$$P=\frac{Kp}{Dif}.$$

### 2.12. Determination of the Concentration Retained in the Pig Vagina

After the permeation studies, the retained D was extracted from the mucosal sections. The mucosa that had come in contact with the formulation was cut into 4 equal pieces (approximately 50 mg each) per Franz cell for introduction into the MagNa Lyser instrument (Roche, Sant Cugat del Valles, Spain) with 1 mL of mobile phase. Homogenization of the tissue was carried out with 5 cycles of 90 s each at 6500 rpm. Finally, the samples were analyzed using the HPLC method [21].

## 3. Results and Discussion

### 3.1. Antifungal Susceptibility Testing

The susceptibility of different species of *Candida* to D was determined in vitro by the microdilution broth method after 48 h of treatment in two different media. The results are presented in Table 4.

RPMI 1640 with MOPS is the medium proposed by CLSI, but the suitability of this medium for D is not well known. To determine if RPMI is adequate, we compared the results with this medium with the results with Sabouraud dextrose, a medium classically used for yeast testing [38,39]. In general, good agreement was obtained between the MIC values with the two different media, confirming the suitability of RPMI 1640.

Regarding the MIC values obtained with RPMI 1640, D was effective against the three species tested at concentrations ranging from 2 to 8 µg/mL. These results correlate well with published data, reporting MIC values ranging from 2 to 8 µg/mL for *C. albicans* (ATCC 90028 and 36082), 16 µg/mL for *C. parapsilosis* (ATCC 22019), and 4 µg/mL for *N. glabrata* (ATCC 90030) [40,41]. The differences in the results could be due to the different strains used from different isolation sources. Furthermore, conventional antifungal (amphotericin B) was used as the control for the yeast susceptibility method. CLSI guidelines suggest that comparing the MICs of different antifungals should not be based solely on the numerical value but rather on how far the MIC is from the breakpoint [27]. Regarding D, there are no defined clinical breakpoints because there are no published data on natural mutants resistant to it. Therefore, it was difficult to use the classical approach to compare D activity with other antifungals.

The key aspects of the disulfiram mode of action that could explain its antifungal activity can be understood by two effects. First, disulfiram is a chelator agent that can sequester copper and consequently alter various metabolic pathways in cells; and second, D has an affinity for the thiol groups of cysteine residues in several cellular targets [21,22]. D may be considered a potential agent for the treatment of candidiasis. Moreover, it has been noted that D inhibits the activities of ABC drug transport proteins that are associated with antifungal resistance [17]. Therefore, its synergism with other antifungals could be an option, and inhibition of ATP pumps would probably result in an increase in the intracellular concentration of the second drug, thus making the second drug, such as an azole, more effective in candidiasis therapy. Additional studies should be carried out to examine this possible effect in clinically relevant strains.

### 3.2. Preparation and Statistical Design to Evaluate the Effect of Experimental Variables on the Mucoadhesive Gel Properties

Once the interesting antifungal properties of D were examined, a novel vaginal placebo formulation was developed to increase mucoadhesion with the vaginal epithelium. Mucoadhesion is defined as the interaction between a synthetic or natural polymer and a mucin surface. It can be affected by different factors, including molecular weight, hydrophilicity, crosslinking, swelling, pH, and polymer concentration [42]. Polymer concentration is an important factor, as there is an optimal concentration at which mucoadhesion reaches a maximum [43]. Additionally, polymers affect the mechanical and rheological properties of the formulation; therefore, controlling the concentration of the polymer combination is important to allow the formulation to adapt to its site of application and ensure appropriate efficacy [44].

PEG-90M was selected as the polymer because of its mucoadhesive properties, its biocompatibility, and its approval by different pharmacopoeias. In addition, PEG-90M presents controlled release properties, especially when used in solid oral dosage formulations [45]. Carrageenan is an anionic polymer with sulfate functional groups that can crosslink with monovalent cations (i.e., sodium and potassium) [46,47]. Carrageenan was selected because it can interact with these ions in the vaginal fluid and increase the hydrogel strength and the residence time in situ, avoiding possible loss of the formulation. Both components have been widely used for vaginal administration of other antimicrobial agents [48,49,50,51,52], including antimicrobial properties for carrageenan [53,54]. This combination has not been described and is of great interest due to the different and complementary mucoadhesion mechanisms of both polymers. Furthermore, due to the hydrophobic nature of D, 20% *w*/*w* HPCD was added to the formulation.

To study the effects of the polymers on the formulation properties (viscosity and mucoadhesion), the percentages of PEG-90M and carrageenan were modified between 0.50% *w*/*w* and 1.50% *w*/*w* and 1.00% *w*/*w* and 2.00% *w*/*w*, respectively. As shown in Table 5, thirteen formulation batches were produced, and the viscosity, mucoadhesion, and amount of adhered fluorescein were characterized for each of them. Fluorescein was selected as the model drug, representing the opposite case compared with D because fluorescein is a hydrophilic compound that would be more easily removed from the formulation when SVF passes over. The viscosity and mucoadhesion were measured with and without SVF to evaluate the effect of carrageenan crosslinking.

As shown in Table 5, the addition of SVF reduced the formulation viscosity in most cases, probably due to a dilution effect, and the effect of the crosslinking effect was now observed. In contrast, the addition of SVF increased the mucoadhesion of the formulation, although the dilution effect (seen with viscosity), mucoadhesion, and viscosity under simulated physiological conditions (in the presence of vaginal fluid) were not correlated.

First, the relationships among the different variables were studied by means of a correlation matrix (Figure 3). This study also tried to reduce the variables in the experimental design and remove highly correlated variables to avoid statistical redundancy.

As Figure 3 shows, mucoadhesion and mucoadhesion with SVF presented a linear relationship. Therefore, because the formulation was applied to the vaginal epithelium, which contains vaginal fluid, the mucoadhesion data without SVF were discarded. The same effect was observed with viscosity, but in this case, viscosity without SVF was chosen because viscosity is related to the administration device (syringe or canula), and as previously seen, there was no relationship between the viscosity and mucoadhesion of the formulation. Finally, fluorescein adhesion did not present a linear relationship.

Surface response methodology was applied using Minitab software (with a significance level of α = 0.05 for the level terms) to evaluate the effects of the tested experimental variables (mucoadhesion with SVF, viscosity, and fluorescein adhesion). The fitted response surface model equations are shown in Table 6. Model selection was based on the adjusted R^2^, which takes into account the different parameters of the nested models. The high R^2^ value of the final equation indicates that it described most of the experimental variability [55].

The amounts of PEG-90M and carrageenan were significant with respect to mucoadhesion with SVF (*p* = 0.017 and *p* = 0.003, respectively). By increasing the amount of polymer, the mucoadhesion increased (Figure 4), as expected. As more polymer was included in the formulation, there were more interactions with mucin, probably due to the formation of hydrogen bonds. The coefficient X_2_ is higher than X_1_, as shown in Table 6, so the effect of carrageenan on mucoadhesion is more important than that of PEG, probably due to the crosslinking effect with monovalent ions in SVF.

Regarding viscosity, a similar effect was observed. The levels of PEG-90M and carrageenan and the interaction between the two variables were significant (*p* = 0.000, *p* = 0.000, and *p* = 0.006, respectively). By increasing the polymer concentrations, the viscosity increased (Figure 5A). In this case, the concentration of PEG-90M had a more significant effect on the viscosity values. Both PEG-90M and carrageenan can form hydrogen bonds; so, the interaction between them influences viscosity. As Figure 5B shows, even though the interaction between the polymers is significant, the viscosity is proportional to the range of polymer concentrations used.

In the case of fluorescein adhesion, a quadratic relationship was obtained (Figure 6A), and a significant interaction between the studied variables was found (Figure 6B). There is a maximum level of adherence (expressed as the fluorescein remaining in the formulation) that decreases at high concentrations of polymer. There is an optimal concentration at which mucoadhesion is maximized. This observed effect may be due to diffusion theory; when the concentration of polymer is high (especially carrageenan because it has a larger spatial conformation than PEG-90M), the interpenetration of the polymeric chains with mucin is hindered, establishing weaker mucoadhesion.

### 3.3. Optimization of the Mucoadhesive Gels

Considering the high R^2^ values from the obtained model, statistical optimization of the formulation was carried out to obtain maximum mucoadhesion and fluorescein adhesion effect and minimal viscosity (to facilitate administration). To obtain maximum mucoadhesion and minimum viscosity (to facilitate syringeability), within the design space limits, the independent variables were optimized. As shown in Table 7, to obtain a mucoadhesion value of >70 mN/cm^2^, viscosity of 33,824 cP, and fluorescein adherence of 98.42%, it is necessary to use 0.866% PEG-90M and 2% carrageenan.

The quality by design study was carried out with the placebo. To check if the inclusion of the drug affected the dependent variables, three additional batches (placebo, 0.5%, and 1% D gel) were produced with the optimized formula (0.866% PEG-90M and 2% carrageenan). The results, as well as model predictability and bias from the theoretical predictions, are reported in Table 8. All obtained results were in the 95% confidence interval (CI), and the bias was <10% in all cases.

### 3.4. Rheological Properties

As there are small differences in viscosity between the optimized placebo formula and the formulations with D, the following characterization was carried out with the placebo. The rheological behavior was studied with and without SVF (Figure 7).

As Figure 7 shows, the formulation presented thixotropy (area of 29.56 Pa/s), and thus the system required more time to recover its original internal structure. The same characteristics were observed for the placebo gel with SVF but with a lower thixotropic area (17.95 Pa/s). Additionally, the viscosity values under the experimental conditions were 2212 ± 10.22 mPa×s and 1586 ± 2.75 mPa×s with and without SVF, respectively. The reduction in viscosity and apparent thixotropy were probably due to the dilution effect of SVF.

Figure 7 shows the possible pseudoplastic behavior of the tested formulation (with and without SVF) due to the apparent viscosity decrease with an increasing shear rate. To confirm the rheological behavior, the experimental data were fit to different equations, as shown in Table 9. A lower Chi^2^ and a higher correlation factor represent the best fit. The model that best fits the experimental data is the Cross equation, which is usually employed to describe the rheological behavior of pseudoplastic materials with yield stress values [56].

The Cross equation is a versatile equation that can estimate different models that converge into simpler models based on different assumptions. Table 10 shows the Cross equation parameters, where for shear-thinning materials, the value of *n* is between zero and one. In this case, the *n* values of the gels with and without SVF were lower than one, which confirms the pseudoplastic profile [56]. The placebo gel without SVF showed a higher viscosity at zero shear than with SVF due to the dilution effect of SVF. In contrast, the *η_∞_* in both cases were very similar, indicating that the gels converged toward similar viscosities at high flow rates with extremely small values compared to the viscosities described previously (2212 and 1586 mPa × s). According to $\dot{ɣ}$ value, the placebo gels without SVF were less shear-thinning than they were with SVF. The rheological flow properties of polymers depend on the structural parameters; in this case, SVF crosslinks with carrageenan, affecting the rheological behavior of the gel.

In addition, a viscoelasticity study was carried out to evaluate the internal structure of the formulation and the effect of SVF on it. Figure 8 and Table 11 show the viscoelasticity parameters (storage modulus G′ and loss modulus G″).

The formulations behaved as solid-like products (G′ > G″), and this value was confirmed by tan(δ) < 1. The loss tangent (tan δ) is a measure of the energy lost to stored energy during cyclic deformation (tan δ = G″/G′). A value of tan δ < 1 indicates prevalent elastic behavior [57]. In this case, the same value of tan δ was obtained for both formulations, and it was <1. The values of both the storage and complex moduli increased in the presence of SVF. When SVF was added to the gel, G″ remained essentially the same as that without SVF, but G′ increased. Carrageenan crosslinking with the monovalent ions of SVF forms a more rigid structure, and therefore, the storage modulus is higher than that without SVF. Despite the increasing values with SVF, the proportion was maintained, and therefore, the value of δ was the same in both cases.

Under an external force, molecular chain orientation is caused by internal friction. When the molecular weight is lower than a certain value, there is a crossover point between the G″ and G″ curves, which means that there is a balance between the states of internal friction and disorientation. With an increase in molecular weight, the crossover point moves to a lower frequency because of restricted disorientation. This lower frequency allows sufficient time for molecular orientation. Therefore, the higher molecular weight molecules need more time for molecular orientation. In summary, the crossover point data demonstrate that the molecular weight decreases with an increase in frequency and polydispersity increases with a decrease in frequency [58]. In this case, as Figure 8 shows, the crossing point is higher without SVF (approximately τ = 140 Pa) than with SVF (approximately 75 Pa). With SVF, sodium ions intercalated into the carrageenan chains, increasing their rigidity and producing a more restricted orientation, so the value of τ was lower.

### 3.5. Syringeability of the Formulations

Syringeability (the force required to extrude a formulation though a syringe) is another important parameter for practical administration with a syringe [50]. In this case, the time needed to empty the syringe was determined with the placebo formulation, which was 3.04 ± 0.0058 s.

The time and force needed to empty the syringe were acceptable and would allow for the easy application of the product to the vaginal area.

### 3.6. In Vitro Release of Disulfiram from the Gel

The release of disulfiram from the 0.5% and 1% mucoadhesive gels is depicted in Figure 9. After 6 h, 80% of D was released from the 0.5% gel, while more than 90% was released from the 1% gel.

After fitting the mean release data to the different mathematical models (Table 12), it was observed that the first-order model provided the best fit (AIC 30.16, adj R^2^ 99.78%) of the experimental data in the case of the 0.5% (*w*/*w*) gel formulation, whereas the Weibull model best fit the data from the 1% (*w*/*w*) D gel (AIC 55.19, adj R^2^ 97.33%).

Once the mean release kinetic behaviors were determined, the means and standard deviations of the individual release data for both gels were calculated, as reported in Table 12.

The first-order model confirmed that the D release mechanism from the 0.5% (*w*/*w*) gel was diffusion, according to Fick’s Law [33,51]. However, the Weibull model was a nonmechanistic equation. Papadopoulou et al. [51] determined a relationship between the shape parameter β and the release mechanism. In the case of the 1% (*w*/*w*) D gel, the value of β was 0.73, which corresponds to the Fick diffusion release mechanism (β < 0.75). The release from both formulations, despite being explained by different models, present the same release mechanism.

### 3.7. Pig Vagina Permeation

In vivo permeation tests were performed with pig vagina and the 0.5% (*w*/*w*) and 1% (*w*/*w*) D gels (*n* = 4). Figure 10 shows the permeation profiles of the tested formulations, and the permeation parameters are listed in Table 13.

D had a good permeation profile, achieving a steady state within the first hour after administration of the 1% D gel (*T_lag_* = 1.17 h) and within the second hour for the 0.5% D gel (*T_lag_* = 2.28 h). The observed mucosal absorption was probably caused by the favorable physicochemical properties of the compound according to Lipinski’s rule of five [59]: low molecular weight (296 Da), log*p* value of 3.88, melting point under 200 °C (71.5 °C), and fewer than five hydrogen bond donors and acceptors [20]. The partition parameter (P) of the API between the gel and the vaginal mucosa revealed that this is the factor that affects the vaginal permeation of both formulations the most. The *p* value obtained was very similar for the 0.5 and 1% gels because it is an independent factor of the D concentration. In this case, the diffusion coefficient had a low contribution to D permeability, and the concentration effect was more noticeable for the 1% formulation according to Fick’s Law. Even though D permeates, as previously noted, D has classically been used orally as a treatment for alcoholism and has been reported to have a good safety profile during clinical use.

In other permeation studies involving human and pig skin, D showed a lack of permeability [21,22]. These differences are due to the nonkeratinization of the vaginal epithelium, and the increase in surface area provided by the transversal rugae of the mucosa would also increase drug absorption across this tissue [60]. Generally, porcine vaginal tissue seems to be a good in vivo permeability model for extrapolation to human vaginal tissue due to substantial histological similarities (stratified squamous epithelium supported by connective tissue) [61,62,63].

### 3.8. Disulfiram Concentration Retained in the Pig Vagina

*Candida* affects the vaginal epithelium and has the capacity to adhere to epithelial cells and penetrate into the tissue. Especially important, in recurrent candidiasis, vaginal relapse following incomplete organism eradication after treatment is a mechanism by which chronification occurs. Furthermore, colonization of the vaginal epithelium facilitates the formation of a yeast reservoir or a source for future infections [8]. Given that our target is the vaginal epithelium, the drug content in the vagina was determined. After permeation studies, the tissue was cut into 50 mg pieces, and these pieces were homogenized with mobile phase using a MagNa Lyser instrument. After HPLC analysis, the mean amount of drug detected in the tissue after administration of the 0.5% D gel was 36.48 ± 2.50 µg and was 87.73 ± 3.12 µg for the 1% D gel. Considering the vaginal density (1 g/mL due to the high water content), the mean D concentrations obtained were 138.17 µg/mL and 268.67 µg/mL, respectively. These concentrations are still higher than the MIC values (2–8 µg/mL), confirming the suitability of D for treating vaginal infections, including those in the deeper mucosal layers or recurrent cases.

## 4. Conclusions

Disulfiram was revealed to be effective against different *Candida* species and is a potential alternative to classic antifungals for the treatment of fungal infections. In this study, disulfiram mucoadhesive gels were prepared successfully for the treatment of vaginal candidiasis. Formula optimization has allowed the development of gels with suitable physicochemical characteristics that facilitate vaginal administration, improving the residence time, patient compliance, appearance of resistance, and typical limitations of conventional vaginal drug delivery systems and traditional antifungals. Furthermore, after gel characterization, the optimized formulation was shown to have appropriate adhesiveness, syringeability, and mucoadhesiveness. Additionally, the rheological behavior revealed the pseudoplastic flow of the gels. The developed gel showed promising in vitro release, with more than 70% release of it in the first 5 h. According to in vitro distribution studies, the drug concentration in vaginal tissue was higher than the MIC value, making D potentially effective for the treatment of candidiasis. However, the clinical implementation of D has encountered some challenges. Further research should be carried out, and an exploration of the molecular mechanism of disulfiram as an antifungal agent is needed. Taken together, these results suggest that disulfiram mucoadhesive gels could be a good option as an alternative treatment for vaginal candidiasis.

## Figures and Tables

**Figure 1 pharmaceutics-15-01436-f001:**
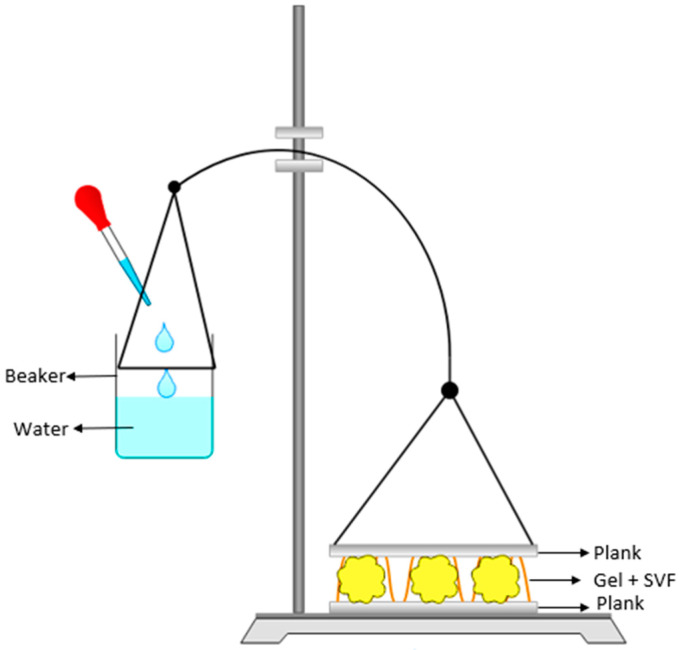
Schematic illustration of the device for evaluating adhesive strength.

**Figure 2 pharmaceutics-15-01436-f002:**
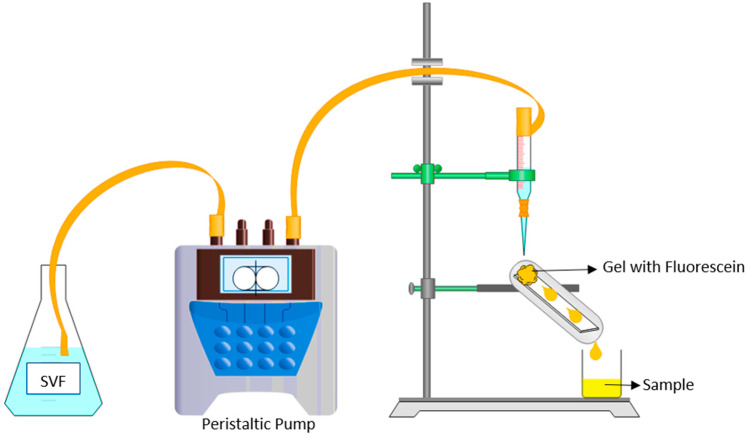
Schematic of the device used to measure the mucoadhesive properties.

**Figure 3 pharmaceutics-15-01436-f003:**
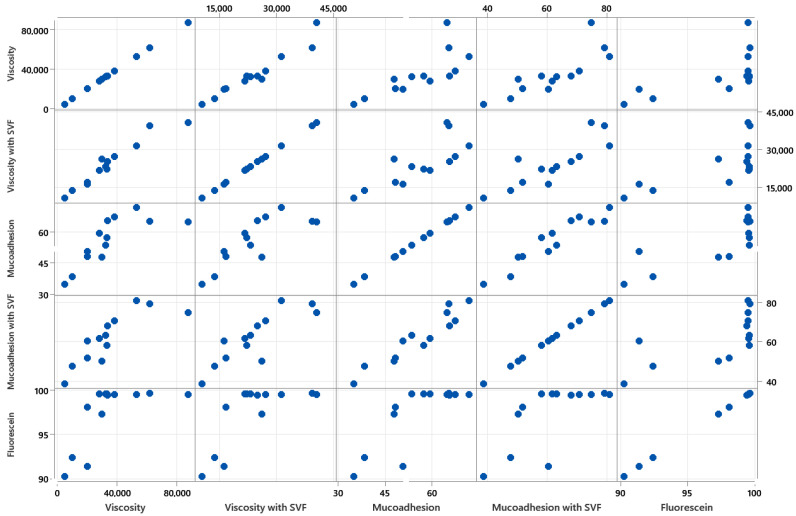
Matrix plots explaining the relationships between the different variables.

**Figure 4 pharmaceutics-15-01436-f004:**
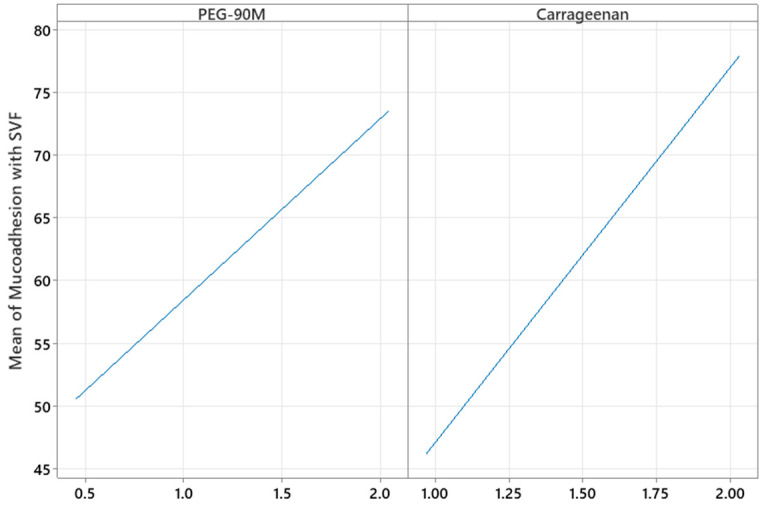
Plots of the main effects of PEG-90M and carrageenan on the mucoadhesive properties of the gels.

**Figure 5 pharmaceutics-15-01436-f005:**
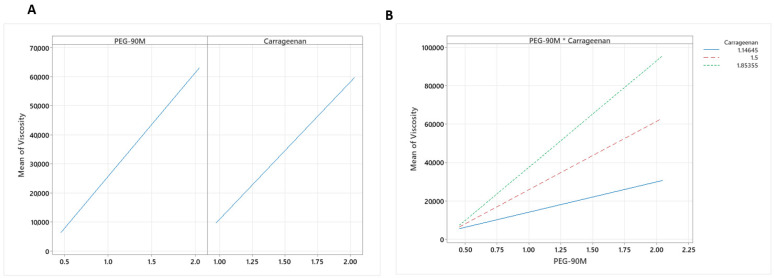
(**A**) Plots of the main effects of PEG-90M and carrageenan on the viscosity (mPa×s) of the gels. (**B**) Viscosity interaction point.

**Figure 6 pharmaceutics-15-01436-f006:**
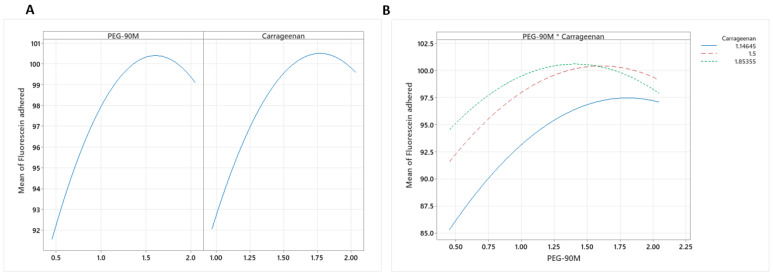
(**A**) Plots of the main effects of PEG-90M and carrageenan on fluorescein adhesion. (**B**) Fluorescein adhesion interaction plot.

**Figure 7 pharmaceutics-15-01436-f007:**
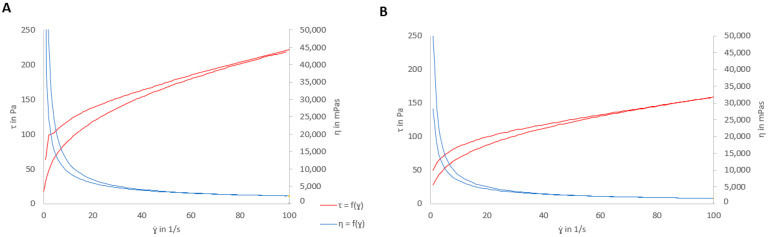
(**A**) Shear stress vs. shear rate (red) and viscosity vs. shear rate (blue) curves of the placebo formulation. (**B**) Shear stress vs. shear rate (red) and viscosity vs. shear rate (blue) curves of the placebo formulation with SVF.

**Figure 8 pharmaceutics-15-01436-f008:**
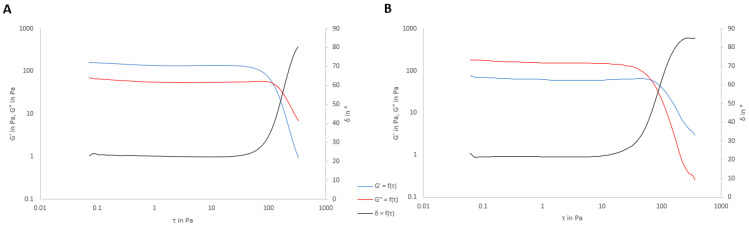
Storage modulus, loss modulus, and phase angle during the sweep stress test on the mucoadhesive gel 24 h after preparation at a frequency of 1 s^−1^. (**A**) Placebo Gel. (**B**) Placebo Gel with SVF.

**Figure 9 pharmaceutics-15-01436-f009:**
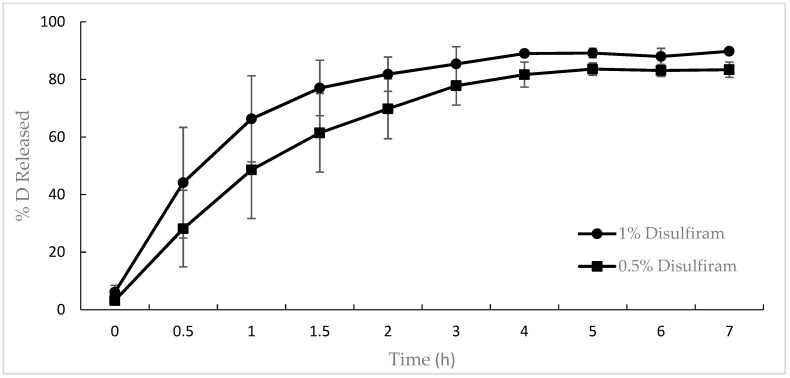
Release of disulfiram from the 0.5% gel and from the 1% gel. Mean of the quantities released of D expressed as a percentage respect total amount seeded from the 0.5% gel and from the 1% gel after 6 h in vitro.

**Figure 10 pharmaceutics-15-01436-f010:**
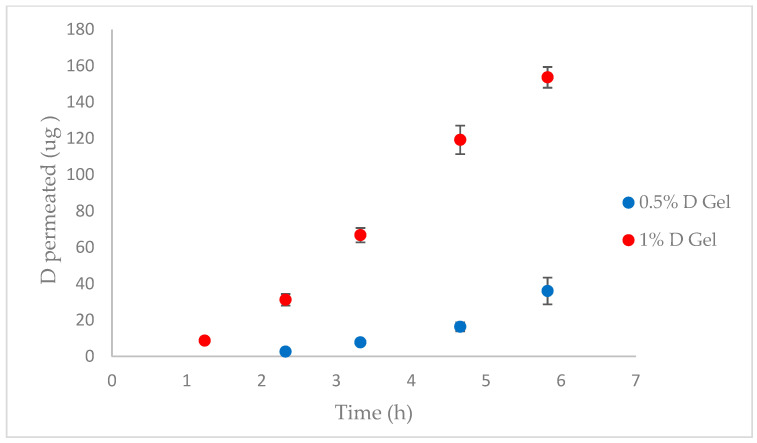
Pig vagina permeation profiles after administration of the 0.5% and 1% D gels (*n* = 4).

**Table 1 pharmaceutics-15-01436-t001:** Levels studied for surface response design.

Factor	Lower Level	Higher Level
% PEG-90M (*w*/*w*)	0.5	1.5
% Carrageenan (*w*/*w*)	1.0	2.0

**Table 2 pharmaceutics-15-01436-t002:** Rheological equations used to evaluate the prepared formulations.

Rheological Model	Equation
Newton	$\tau =\eta \times \dot{\gamma }$
Bingham	$\tau =\tau_{0}+(\eta_{0}\times \dot{\gamma })$
Ostwald–de Waele	${\tau =K\times \dot{\gamma }}^{n}$
Herschel–Bulkley	$\tau =\tau_{0}+{K\times \dot{\gamma }}^{n}$
Casson	$\tau =\sqrt[{n}]{\left(\tau_{0}^{n}+{\left(\eta_{0}\times \dot{\gamma }\right)}^{n}\right)}$
Cross	${{\tau =\dot{\gamma }\times {(\eta }_{\infty }+{(\eta }_{0}-\eta_{\infty })/(1+(\dot{\gamma }/\dot{\gamma }}_{0})}^{n})$

τ: Shear stress (Pa), η: viscosity (cP or Pa-s), $\dot{\gamma }$: shear rate (s^−1^), $\tau_{0}$: minimum yield stress (Pa), $\eta_{0}$: zero shear viscosity (cP or Pa-s), *K*: consistency factor (Pa s^n^), $\eta_{\infty }$: infinite shear viscosity (cP or Pa-s), ${\dot{\gamma }}_{0}$: zero shear rate (s^−1^).

**Table 3 pharmaceutics-15-01436-t003:** Different kinetic models and equations tested.

Kinetic Model	Equation	Parameter(s)
First-Order	$F=F_{max}\left({1-e}^{\left({-K}_{1}t\right)}\right)$	$F_{max}$, *K*_1_
Higuchi	$F=K_{H}\times t^{1/2}$	*K* _ *H* _
Korsmeyer–Peppas	*F* = *K*_*KP*_ × *tn*	*K*_*KP*_, *n*
Weibull	F=Fmax×1−e(−tTd)β	$F_{max}$, *α*, *β*

*F*: Fraction of drug released, *t*: time, $F_{max}$: maximum fraction of drug released, *K*_1_: first-order constant, $K_{H}$: Higuchi constant, *K*_*KP*_: Korsmeyer–Peppas constant, *n*: diffusional exponent, *T_d_*: the time at which 63.2% of the drug had been released, *β*: Weibull shape parameter.

**Table 4 pharmaceutics-15-01436-t004:** Summary of the MICs (*n* = 3) for disulfiram with three different species of *Candida* using two different test media.

	MIC (µg/mL)	
	Test Media	
*Candida* spp.	RPMI 1640	Sabouraud Dextrose
*C. albicans*	2	2
*C. glabrata*	4	2
*C. parapsilosis*	8	8

**Table 5 pharmaceutics-15-01436-t005:** Influences of PEG-90M and carrageenan on the physicochemical parameters.

Batch	PEG-90M (% *w*/*w*)	Carrageenan (% *w*/*w*)	Viscosity (cP)	Viscosity with SVF (cP)	Mucoadhesion (mN/cm^2^)	Mucoadhesion with SVF (mN/cm^2^)	Adhered Fluorescein (%)
LP-83	0.50	1.50	19,910	16,170	50.6	60.5	91.40
LP-84	2.00	1.50	61,660	39,350	65.3	79.3	99.64
LP-86	1.25	2.00	52,840	31,360	71.9	80.9	99.50
LP-87	1.25	1.50	38,000	27,070	67.3	70.8	99.52
LP-88	1.25	1.50	33,170	22,210	57.4	58.1	99.60
LP-89	1.25	1.50	27,890	21,730	59.4	61.8	99.59
LP-90	1.25	1.50	33,370	25,070	65.5	68.2	99.43
LP-91	1.25	1.50	32,260	23,250	53.6	63.3	99.60
LP-92	1.78	1.14	29,750	26,180	47.9	50.3	97.33
LP-93	0.72	1.85	20,100	16,790	48.3	51.9	98.10
LP-94	0.72	1.14	4720	10,490	34.9	38.8	90.28
LP-95	1.78	1.85	87,280	40,590	64.8	74.9	99.54
LP-96	1.25	1.00	10,000	13,690	38.4	47.8	92.42

**Table 6 pharmaceutics-15-01436-t006:** Response surface model equations for monitoring the effect of polymer concentration on the properties of mucoadhesive gels.

Quadratic Polynomial Model Equation	R^2^	Adj R^2^
Y_1_ = −0.8 + 14.40 X_1_ + 29.88 X_2_	0.702	0.642
Y_2_ = 24,704 − 48,649 X_1_ − 23,052 X_2_ + 56,200 X_1_ X_2_	0.945	0.927
Y_3_ = 28.85 + 32.60 X_1_ + 56.06 X_2_ − 6.652 X_12_ − 13.21 X_22_ − 7.48 X_1_X_2_	0.987	0.978

X_1_: Concentration (% *w*/*w*) of PEG-90M; X_2_: concentration (% *w*/*w*) of carrageenan. Y_1_: mucoadhesion with SVF (mN/cm^2^), Y_2_: Viscosity (cPs), Y_3_: fluorescein adhered (%).

**Table 7 pharmaceutics-15-01436-t007:** Prediction of polymer concentrations for a formulation with maximum mucoadhesion and fluorescein adhesion and minimal viscosity.

PEG-90M	Carrageenan	Fluorescein Adhesion Fit	Mucoadhesion with SVF Fit	Viscosity Fit	Composite Desirability
0.866	2	98.42	71.46	33,824	0.76

**Table 8 pharmaceutics-15-01436-t008:** Results after characterization of the final formulations with bias with respect to the theoretical value.

Property	Theoretical Value (95% CI)	Placebo Gel	0.5% D Gel	1% D Gel	Bias (%)
Mucoadhesion (mN/cm^2^)	71.5 (60.85; 82.07)	72.9	68.2	69.4	1.86
Viscosity (cP)	33,824 (22,885; 44,763)	29,835	30,400	32,780	8.30
Fluorescein adhered (%)	98.42 (97.23; 99.62)	98.67	98.58	98.81	0.27

**Table 9 pharmaceutics-15-01436-t009:** Rheological models with Chi^2^ and r values for the placebo gel and placebo gel with SVF.

Rheological Model	Placebo Gel		Placebo Gel with SVF	
	Chi²	r	Chi²	r
Newton	1.773 × 10^5^	0.326	1.004 × 10^5^	−0.137
Bingham	1.147 × 10^4^	0.970	6127	0.968
Ostwald–de Waele	8.997	1.000	13.18	0.999
Herschel–Bulkley	7.020	1.000	12.900	0.999
Casson	9.220	0.988	1791	0.991
Cross	1.164	1.000	1.359	1.000

**Table 10 pharmaceutics-15-01436-t010:** Rheological model fitting of the placebo gels with and without SVF.

Cross Equation Parameter	Placebo Gel	Placebo Gel with SVF
η₀	318.7	141.9
$\eta_{\infty }$	0.1517	0.1600
$\dot{ɣ}$	0.0422	0.1235
N	0.6488	0.6853

**Table 11 pharmaceutics-15-01436-t011:** Viscoelasticity parameters of the Placebo Gel and Placebo Gel with SVF (mean ± standard deviation (SD)).

Parameter	Formulation	Mean ± SD
G′	Placebo gel	129.92 ± 25.20 Pa
	Placebo gel with SVF	149.66 ± 25.33 Pa
G″	Placebo gel	57.90 ± 4.60 Pa
	Placebo gel with SVF	63.67 ± 4.01 Pa
η*	Placebo gel	22,709 ± 3657 mPa
	Placebo gel with SVF	25,947 ± 3649 mPa
δ	Placebo gel	24.20 ± 4.46°
	Placebo gel with SVF	23.67 ± 5.07°
tan(δ)	Placebo gel	0.45 ± 0.11
	Placebo gel with SVF	0.44 ± 0.08

**Table 12 pharmaceutics-15-01436-t012:** Individual models and parameters for the release of D from the 0.5 and 1% gels.

Formulation	Model	Parameters	Value
0.5% D gel	First-Order	*K*_1_ (h^−1^) *F_max_* (%)	0.95 ± 0.48 85.79 ± 2.13
1% D gel	Weibull	α β *F_max_* (%)	0.88 ± 0.43 0.73 ± 0.23 93.42 ± 3.21

**Table 13 pharmaceutics-15-01436-t013:** Mean parameters of D permeation after gel application to pig vagina.

Formulation	Parameter	Mean	SD
0.5% D gel	*Jsup* (µg/h·cm^2^)	9.0106	0.6562
	R^2^	0.9113	0.0445
	*Kp* (cm/h)	0.2481	0.0181
	*T_lag_* (h)	2.2834	0.6721
	*P* (cm/h^2^)	3.4376	1.2485
	*Dif* (1/h)	0.0763	0.0225
1% D gel	*Jsup* (µg/h·cm^2^)	32.9769	1.1737
	R^2^	0.9913	0.0001
	*Kp* (cm/h)	0.4542	0.0162
	*T_lag_* (h)	1.1698	0.0514
	*P* (cm/h^2^)	3.1851	0.0265
	*Dif* (1/h)	0.1426	0.0063

## Data Availability

Data are available upon request due to intellectual property.

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
