# Peer review of "Repurposing Disulfiram as an Antifungal Agent: Development of a New Disulfiram Vaginal Mucoadhesive Gel"

_pharmaceutics, 2023, doi:10.3390/pharmaceutics15051436_

Round 1

Reviewer 1 Report

The submitted manuscript deals with the development and evaluation of a novel vaginal hydrogel containing disulfiram for the treatment of vulvovaginal candidiasis. Investigation of new approaches for treating vaginal infections is of great importance particularly due to the increasing trend of microbial resistance. The manuscript is generally well-written and fits under the scope of the Journal.  There are just a few comments that should be considered to strengthen the manuscript.

Although the physicochemical properties of disulfiram are mentioned in Results and discussion, they should be also provided (would better fit) in Introduction to improve the clarity of the experimental setup and expected pharmacokinetic behavior i.e., solubility, log P and Mw.

Shortly explain/write in the manuscript the reason(s) for selection buffer pH 5.5 in in vitro release studies, and why were the studies performed at 32 °C? Why not 37 °C?  32 °C corresponds to skin temperature.

Provide the thickness of the vaginal mucous membrane used in permeation studies. The thickness of membrane significantly affects the permeation. How did you prepare the samples of vaginal tissue?

Since disulfiram is not a classical antifungal drug, why haven' you tested conventional antifungal drug as control in antimicrobial studies? This should be mentioned in the manuscript.

Figure 9. Error bars should be in both directions (plus and minus). Re-write the caption for fig 9. to „Release of disulfiram from the 0.5% gel and from the 1% gel“. Indicate that the data are mean values of ? measurements. Y-axis should be entitled to Disulfiram, not % release.

Please be consistent with using abbreviations in the manuscript. Check all the abbreviations and correct, where appropriate. Provide the list of abbreviations.

Line 590: ...“ pig vaginal skin“?? re-write.

In vitro, in vivo – should be in italic.

Author Response

Please, find the document attached. 

Thank you, 

Maria 

Reviewer 2 Report

2  

8 My review 

1

Author Response

Please find the docuemnt attached. 

Thank you, 

Maria 

Reviewer 3 Report

The authors focus on the activity of disulfiram and an innovative delivery method, to treat vulvovaginal candidiasis, a condition caused by the yeast Candida spp. This condition can re-occur frequently, and antifungal resistance is on the rise, which complicates the treatment of this illness. In their study, the authors revealed that disulfiram is effective against different Candida species, and formulate it into interesting mucoadhesive gels that are very extensively characterized. I congratulate the authors for a very interesting work that is very-well conducted.

Author Response

Dear Sir/Madame

Thank you for providing your valuable feedback on my manuscript

Maria 

Reviewer 4 Report

Abstract:

This is concise and well-written. It has provided a good summary of the article. The authors should be aware of the recent update of the fungal taxonomy for yeasts. For instance, Nakaseomyces glabrata for C. glabrata and Pichia kudriavzevii for C. krusei.

Introduction:

The authors have provided a good summary of different yeast species and their antifungal resistance profiles. The available treatment options, the mechanism of action of Disulfiram, and the aim of the work were noted

Materials and Methods:

The section is well-written and reproducible.

Conclusions:

The paper will attract a wide readership, especially in the era of promoting good antifungal stewardship.

The article should be accepted after the correction of the new names due to the change in Taxonomy.  

What is the main question addressed by the research?

The main questions addressed by the authors was to look at the potential ability of the alternative formulations that could be used against vulvovaginal candidiasis. Mucoadhesive gels with disulfiram offers those potential benefits. The aim of the study was to develop and optimize a mucoadhesive delivery system for the local administration of the disulfiram.

Do you consider the topic original or relevant in the field?

The topic is original and it is relevant in the field of mycology as the Candida species are increasingly becoming resistant to conventional antifungals. The results will definitely address ta specific gap in the field of mycology.

What does it add to the subject area compared with other published material?

It has added value to the existing published article in the area of antifungals with intravaginal delivery routes.

What specific improvements should the authors consider regarding the

methodology? What further controls should be considered?

 The methodology is considered comprehensive enough and its reproducible.

Are the conclusions consistent with the evidence and arguments presented?

and do they address the main question posed?

The conclusion addressed the evidence provided and it has contributed to the knowledge of antifungal agents

Are the references appropriate?

The references are adequate and appropriate for the article

Please include any additional comments on the tables and figures.

Tables and figures are adequate 

Author Response

Dear Sir/Madame

Thank you for providing your valuable feedback on my manuscript. We agree with your comments, and we have been able to incorporate changes to reflect your suggestions provided. 

Maria 

Round 2

Reviewer 2 Report

There are several places in the manuscript where C. glabrata is changed to N. glabrata.  I do not understand this change. The correct genus name is Candida.  

Author Response

Dear reviewer, 

We have changed the name of C. glabrata according to the suggestion of reviewer 4: "The authors should be aware of the recent update of the fungal taxonomy for yeasts. For instance, Nakaseomyces glabrata for C. glabrata and Pichia kudriavzevii for C. krusei". 

https://www.ncbi.nlm.nih.gov/Taxonomy/Browser/wwwtax.cgi?mode=info&id=5478

https://www.ncbi.nlm.nih.gov/Taxonomy/Browser/wwwtax.cgi?mode=info&id=4909

However, we keep C. glabrata because it is an accepted synonym (new version of the manuscript is attached). 

Thanks you, 

Maria